# In Vitro Effect of Modified Polyetheretherketone (PEEK) Implant Abutments on Human Gingival Epithelial Keratinocytes Migration and Proliferation

**DOI:** 10.3390/ma12091401

**Published:** 2019-04-29

**Authors:** Liza L. Ramenzoni, Thomas Attin, Patrick R. Schmidlin

**Affiliations:** 1Clinic of Preventive Dentistry, Periodontology and Cariology, Center of Dental Medicine, University of Zurich, Plattenstrasse 11, 8032 Zurich, Switzerland; thomas.attin@zzm.uzh.ch (T.A.); patrick.schmidlin@zzm.uzh.ch (P.R.S.); 2Laboratory of Applied Periodontal and Peri-implantitis Sciences, Clinic of Preventive Dentistry, Periodontology and Cariology, Center of Dental Medicine, University of Zurich, Plattenstrasse 11, 8032 Zurich, Switzerland

**Keywords:** keratinocytes, implant abutments, PEEK, zirconia, titanium

## Abstract

Improving soft tissue attachment to implant abutments is a crucial factor for enduring health and maintenance of soft peri-implant tissue health. In this in vitro study we aimed to compare the biocompatibility of three different abutment surfaces: titanium, zirconia and modified polyetheretherketone (PEEK). Surface topography, roughness and wettability were investigated with scanning electron microscopy, profilometer and contact angle meter, respectively. Human gingival epithelial keratinocytes were examined for viability, morphology, proliferation and migration by using tetrazolium salt colorimetric assay, scanning electron microscopy imaging, immunofluorescence bromodeoxyuridine analysis and scratch wound healing assays. Roughness measurements revealed differences between the investigated surfaces. Keratinocytes cultured on all examined surfaces indicated adhesion and attachment by means of scanning electron microscopy imaging. Cell viability assays showed no significant differences between the groups (*p* > 0.05). The modified PEEK surface similarly improved surface roughness in comparison to titanium and zirconia, which resulted in greater and equivalent cell proliferation and migration. The study methodology showed here may emphasize the importance of cell interactions with different abutment materials, which in part increases the changes of implant success. PEEK, titanium and zirconia surface types used in this study showed mostly similar epithelial biological responses.

## 1. Introduction

A central issue in long-term dental implant success is not only closely related to the integrity of osseointegration, but also epithelium health and the quality of attachment of the connective tissue to the implant abutment surface [1,2]. A variety of bacterial species can penetrate the transmucosal tissue surrounding the dental implant, whereby these tissues should ideally act as barrier between the oral environment/bacterial infection and peri-implant bone [2]. Also, good marginal fit between implant and abutment is important to prevent the bacterial microleakage that may interfere with peri-implant tissue health [3]. Numerous experimental studies with different cell types have been used to analyze cell behavior response towards biomaterials and dental implant abutment surfaces [4,5,6]. Many studies have therefore also focused on human gingival epithelial keratinocytes (HGEK) as important cells associated with soft tissue interaction with implant attachment and lining [7,8,9,10].

Besides meeting esthetic and functional requirements, abutments need to provide adequate adaptive tissue responses [11,12,13]. Titanium has been predominantly used for clinical abutments so far and has proven to be the material of choice due to its long-term success and biocompatibility. However, titanium abutments in the anterior region often lead to poorer esthetics even when subgingivally located in anterior region. Thus, as an alternative to titanium, zirconia -based implant abutments have also been introduced as a valuable option for patients with high-esthetic expectations [14,15]. Studies with various cell types on biocompatibility have compared titanium implant to glass ceramics and zirconia. The results demonstrated significant differences concerning in cell viability and migration ability of gingival fibroblasts and oral keratinocytes [14,16]. Another well-known type of implant abutment is made of a high performance thermoplastic polymer called polyetheretherketone (PEEK), which is in the meanwhile considered as an alternative biomaterial to metallic implant, as PEEK neither release ions, forms by-products, nor corrode or degrade [17,18,19,20]. 

In order to better understand the phenomenological behavior of cells of oral soft-tissue cells around different implant abutments, more knowledge is still required. Methods of improvement of surface nanostructure and its biologic response has been extensively investigated in recent years [21]. In order to prevail over implant failure, possible bacterial adhesion and infiltrations or even inadequacy of the implant surface for biocompatibility, should be avoided. In this sense, bioengineering of implant nanostructures has significantly advanced the development of new biomaterial [22,23]. Nanostructure surfaces patterns on oxidation and anatase has been, for example, proposed to increase cell activity and biological response of the healing site. Furthermore, these surfaces were shown to in part increase antimicrobial properties [23]. Other studies have been conducted to investigate the physical and chemical properties of PEEK as bioinert implant abutment material [24,25,26,27,28]. However, limited information is available regarding the cell interactions and consequent soft tissue integration. In this aspect, the present study may have significant implications on the clinical field of regenerative implant medicine, as it draws attention to the importance of surface improvement to favor implant-tissue healing and integration.

The implant abutment profile may have implications on the establishment of surrounding epithelial attachment leading to necessary initial healing and physiologic gingival contour. Anticipated long-term implant health and success may also be influenced by color aesthetics of the material and dental implant restorations. PEEK can be considered as a biologically satisfactory material for implant abutments, however, additional in vitro cellular studies are needed for a complete understanding of the biological performance of modified PEEK abutments over a long time period in relationship to oral tissues. Additionally, there is a lack of literature concerning the influence of modified PEEK abutments on oral gingival keratinocytes cell proliferation and wound healing response in comparison to titanium and zirconia abutments. 

Therefore, the aim of the present in vitro study was to characterize the cell response of human gingival epithelial keratinocytes cultured (HGEK) on modified PEEK abutment surface in comparison to titanium (Ti) and zirconia (ZrO_2_) abutments. We hypothesized that the type of abutment material could influence the immediate abutment-tissue interaction on a cellular level and may play a key biocompatibility function in oral soft-tissue formation around implant abutments. PEEK may be an equivalent, or even better, alternative material in terms of biocompatibility for soft-tissue cells, which could influence peri-implant tissue formation and ultimately the esthetic outcome after implant placement.

## 2. Materials and Methods

### 2.1. Abutment Material Surface Morphology, Roughness and Wettability Analysis

The ZrO_2_ Computer-Aided-Design/Computer-Aided-Manufacturing (CAD/CAM), uncoated Ti (Ti6A14V-ELI) and modified PEEK discs (12 mm ∅ × 2 mm) were fabricated and provided by one manufacturer (Zimmer Biomet 3i, Palm Beach Gardens, Florida, USA) and compared with uncoated polyester cell culture discs, which were used as controls. The macrostructure, microstructure and nanostructure images of the different materials disc surfaces were obtained by using scanning electron microscopy (SEM, FIB-SEM 40 CrossBeam Zeiss Auriga, Carl Zeiss, Jena, Germany) at 1k× and 100k× magnifications. Surface disc images were recorded with no additional coating and SEM specifications were used under vacuum with 2 mm distance and with 5 kV secondary electron detector. The surface topography was measured by using the Talysurf Intra 50 profilometer (Taylor Hobson, Leicester, UK). Values of surface roughness (R_a_ in μm) were determined using a 0.8 mm cut-off value and 4 mm as the length measurement. Roughness measurements were made in different points in five distinct discs for each material. Surface wettability of each surface type was examined using a contact angle meter (SL200, USA Kino Industry, Norcross, GA, USA). We measured 1 μL drops of deionized water in six different discs per material at three distinct points and after 3 s of water droplets application. The contact angle with water was photographed using Image-Pro Plus version 6.0 (Media Cybernetics Inc., Bethesda, MD, USA). A contact angle closer to 0° was considered hydrophilic and greater than 90° considered hydrophobic. Measurement of each material wettability was a result of average values obtained from the six discs. 

### 2.2. Cell Culture

Immortalized HGEK were previously established (Bao K et al., 2014) and donated by the Oral Microbiology Institute, Center of Dental Medicine, University of Zurich. HGEK cells were cultured in an incubator (5% CO_2_, 95% air at 37 °C) and passaged at regular intervals depending on their growth characteristics using 0.25% trypsin (Seromond Biochrom, Berlin, Germany) and maintained in complete epithelial medium consisting of defined keratinocyte serum free medium (Gibco, Life Technologies GmbH, Carlsruhe, Germany), supplemented with 100 U/mL penicillin (Sigma, St. Louis, Missouri, USA, 15140-122), 100 mg/mL streptomycin (Sigma, St. Louis, Missouri, USA), 2 mM L-glutamine (Sigma, St. Louis, Missouri, USA, G7513), and 0.25 mg/mL fungizone (Sigma, St. Louis, Missouri, USA). Change in cell culture medium and cell passage were conducted two times in five days using new culture medium. The cells used in this study were between the fifth and fifteenth passage.

### 2.3. Cell Attachment and Morphology

Surfaces were analyzed using scanning electron microscopy (SEM, FIB-SEM 40 CrossBeam, Zeiss Auriga, Carl Zeiss, Jena, Germany) in order to quantify cell attachment. HGEK cells (2 × 10^4^) were seeded cultured in incubator at 37 °C in 5% CO_2_ for 24 h following passage protocols described above. After washing with phosphate buffered saline (PBS 1x) non-adherent cells were aspirated. For the SEM protocol analysis, cells were washed with a 0.1 M PBS (Gibco, Life Technologies GmbH, Carlsruhe, Germany). Then, cells were fixed for 6 h with 4% glutaraldehyde fixative solution (Sigma, St. Louis, Missouri, USA). The excess of fixative solution was removed, before repeated cell washing with PBS. Then, cells were additionally rinsed three times with PBS 1x for 5 min and fixed in OsO_4_ for 15–30 min. Finally, cells were dehydrated through an ethanol bath series starting with 30% and changing to solutions of 50%, 70%, 90%, and three times 100%. Lastly, the samples dried, and a 100 nm thick layer of gold-palladium was used with an ion coater (Eiko IB-type 3). Cells on the discs were finally observed by SEM (FIB-SEM 40 CrossBeam, Zeiss Auriga, Carl Zeiss, Jena, Germany) and images were recorded at 1k× and 100k× magnification.

### 2.4. Cell Viability and Proliferation

To examine the cell viability, HGEK cells were seeded on the different disc materials and tested with a tetrazolium bromide colorimetric assay (MTT; 3-(4,5-dimethylthiazol-2-yl)-2,5-diphenyl, Sigma, Steinheim, Germany). For this experiment, the mid-log HGEK cells ((1 × 10^6^ cells/well, between 3rd and 5th passages) were cultured in 24-well plates with acrylic discs as controls and tested materials. Cell viability was measured after 24 h of cell growth in four different areas [15] with a Synergy HT multi-mode microplate reader (BioTek Instruments, Winooski, VT, USA). The MTT assay was performed in triplicates. To evaluate cell proliferation, the bromodeoxyuridine (BrdU) Kit (Roche Diagnostics GmbH, Mannheim, Germany) was used following protocol instructions. Briefly, HGEK (1.5 × 10^4^ cells/well) were seeded onto 96-well plates and 10 μM of BrdU was added for 24 h. BrdU nucleai incorporation was determined by absorbance at 450 nm.

### 2.5. Cell Migration

Cell migration and motility were evaluated by a scratch wound assay as previously described [29]. HGEK were seeded onto Ti, ZrO_2_ and PEEK disc surfaces at concentration 2 × 10^4^ cells/well in 24-well plates and cultivated under serum starvation and 10 µg/mL mitomycin C to block cell proliferation. In order to avoid cell proliferation during cell migration, serum starvation was employed 16 h before the in vitro making of the “wound” scratch. To reassure no change in cell number, cell counting was performed at beginning and end of the experiment. Next, each disc was artificially wounded by creating a scratch with a plastic pipette 10 μL tip on the cell monolayer. Cells were treated with 0.5 μM fluorescent dye Hoechst 33342 for 30 min in order to create nucleai labeling (Sigma, St. Louis, Missouri, USA). Dye solution was removed and cells were washed with fresh medium (Gibco, Life Technologies GmbH, Carlsruhe, Germany) and cultured for additional 12 h and 24 h. Images of scratch wounds were captured using fluorescence inverted microscope (Zeiss Observer Z1; Intelligent Imaging Innovations) at 0, 12, and 24 h after wounding and wound width measurements were subtracted from wound width at time zero to obtain the net wound closure. Additional experiments were conducted for images recording only at 0, 12, and 24 h. Average width of the scratches at different intervals was determined after cell migration and distance between wound edges was calculated. The wound closure areas were measured with ImageJ (Software 1.48q, Rayne Rasband, National Institutes of Health, USA) after 24 h. In order to avoid scratch width variation, “relative wound closure” area (RWC) was calculated (RWC [%] = wound closure area [pixel] × 100 [%]/× [pixel]. Experiments were repeated at least three times.

### 2.6. Statistical Analysis 

Comparison between individual groups were made using the paired two-tailed t-test. Statistically significant differences in the R_a_ surface values were determined by ANOVA. Differences were considered significant if *p* < 0.05. Statistical analysis was performed using the software SPSS 22.0 software the values were shown for at least three different experiments performed in triplicates (median ± standard deviations).

## 3. Results

### 3.1. Cell Viability and Morphology Analysis

Cell viability analysis indicates that there was only a little and insignificant difference between the materials at 24 h (*p* = 0.1291). The materials presented a similar increase in cell viability activity after 48 h of cell culture; however, no significant differences were observed between the groups (*p* > 0.05). Similar viability degree was observed on the cells over Ti, ZrO_2_ and PEEK surfaces (Figure 1). Images of HGEK cells shape and growth was recorded using SEM for each of the materials (Figure 2). HGEK cell orientation on Ti group and ZrO_2_ discs was in parallel to disc microgroove directions after 14 days, whereas the cells seeded on PEEK discs were rather disposed in random orientation and directions during the same period (Figure 2d–f). Most cells were found present inside the microgrooves with increased filopodia formation. 

### 3.2. Cell Proliferation and Migration

The proliferation assay data showed that PEEK material significantly stimulated the proliferation of HGEK cells after seven days of growth as compared to the control. However, this stimulation effect was found to be similar when compared to Ti and ZrO_2_ materials. No significant differences were observed between the groups (*p* > 0.05, Figure 3). Regarding the cell migration, scratch distances and width closure was obtained by software comparison between images from time 0 to lastly 24 h. We observed that after 12 h, the HGEK cells migrated and covered approximately 50% to 60% of the wound area quantified in time zero for all abutment discs. However, lower migration was observed on Ti and ZrO_2_ when compared to PEEK. Initial wound edges marked initial cell migration and were used to identify the decrease in wound width throughout the experiment. Migration distances were showed separately during periods 0–12 h (migration during first 12 h period) and 12–24 h (during second 12 h period). On the first image recording period after wounding (0–12 h), a significant difference in migration distance was found for PEEK discs when compared to the control, whilst no difference in motility was found on Ti and ZrO_2_ discs (Figure 4, * *p* < 0.01). After 24 h, there was no significant change regarding cell migration between the different materials under investigation. Experiments are showed in triplicates (mean ± standard errors, * *p* < 0.05). 

### 3.3. Abutment Material Surface Characterization

The mean surface roughness values R_a_ (mean ± SD) of the Ti, ZrO_2_ and PEEK discs accounted for 0.086 ± 0.006, 1.352 ± 0.186 and 0.827 ± 0.012 μm respectively. Thereby, the R_a_ values of Ti were significantly lower than those of ZrO_2_ and PEEK specimens (*p* < 0.01 for both comparisons). No significant difference was found between the R_a_ of Zirconia and PEEK specimens (1.152 μm and 0.827 μm; *p* = 0.983) (Table 1). Microscopically, the roughness obtained on the abutment discs consisted of almost parallel longitudinal grooves and ridges (Figure 5). The grooves on Ti, Zirconia, and PEEK were spaced approximately 1.84 ± 1.12, 3.62 ± 2.5, 4.15 ± 1.50 μm apart, respectively, as may be seen in the respective representative SEM micrographs (Figure 5a,c,e). The differences in the distances between grooves were statistically significant at a micro (Figure 5a,c,e) and submicron levels (Figure 5b,d,f). Figure 5a,c,e showed almost similar groove topography. However, viewed under 100k× magnification (Figure 5b,d,f), nanostructures were visible, especially on zirconia and PEEK surfaces (Figure 5e,f). The wettability (median water contact angles) of each specimen was altered between the materials (Table 1). Zirconia and PEEK showed significantly higher contact angles (98.2° and 90.3°) than the hydrophilic Ti surface (72.3°, *p* < 0.01, Figure 5g–i). Surface roughness of the discs was not changed after application of wettability in any of the parameters measured.

## 4. Discussion

The biocompatibility of an implant abutment has a significant clinical relevance, as the material chosen is in intimate and limitless contact with the surrounding soft tissues. The application of a high-performance polymer PEEK abutment in the esthetic zone - at least as a provisional material - is becoming more popular due to its quality and the ability of the material to provide with an appropriate tissue response. In this regard, it has been reported that the modified PEEK abutments may have a positive effect on the cell attachment and proliferation [30,31,32], modified PEEK was selected for this study and compared to well-established titanium and zirconia abutment materials, with regard to their influence on human gingival epithelial keratinocytes. The findings of this in vitro study support the hypothesis that a modified PEEK surface at least similarly increases cell adhesion, viability, and proliferation of gingival keratinocytes to a similar degree as titanium and zirconia. 

Long-term survival of dental implants depends not only on the ability of an abutment material to perform an appropriate host response, but also in part on the control of bacterial infection in the peri-implant region [33]. The microorganism reservoirs found around dental implants and which contribute to implant failure seem to be interchangeable to the ones identified around natural teeth [33]. For this reason, it is important to consider the antimicrobial properties of the material substratum in order to reduce or avoid bacterial adhesion and further surface colonization. Both surface roughness and wettability hold a direct effect on bacterial adhesion [34,35]. Zirconia was shown to have similar colonization potential compared to grade 1 pure titanium [36] and the rougher PEEK seem to present greater bacterial adherence compared to smoother PEEK [37]. However, implant material roughness and wettability may not fundamentally influence adhesion of microorganisms as for the most part may be different between bacterial species [34]. To improve osteoblast response on the surface, special titanium coatings, such as anatase or oxidation patterning, can also be produced around the implant surface and may have additional antibacterial properties [23,34,38]. Thus, further developments in dental implantology should focus on understanding the bacterial and tissue interaction to material surfaces topography/chemistry, which plays an important role on long-term peri-implant health. 

The increase in cell response on PEEK abutment surface was corroborated in previous clinical studies where PEEK was shown to be comparably biocompatible with surrounding tissues similar to titanium [39]. In addition, partially due to its positive esthetics, modified PEEK may be a suitable alternative material to titanium implant abutments. Positive cell response on the modified PEEK surface used in this study may be explained and correlated with its increased wettability [29]. PEEK usually displays greatest changes in contact angle values or wettability after surface treatments [40]. As highly hydrophobic surfaces reduce cellular adhesion, a moderate wettability is normally required to improve interactions of the surface with surrounding tissues. Polymers are commonly known as greatly bioinert hydrophobic materials with low surface energy [41,42]. In fact, a PEEK abutment surface reduces osteoblast differentiation when compared to titanium surfaces. Consequently, PEEK surfaces are usually coated and blended with bioactive particles in order to increase osteoconductive properties [43,44]. Other modifications were developed in an attempt to enhance surface properties and biocompatibility of PEEK materials. In particular, titanium dioxide and hydroxyapatite bioactive nanoparticles were added to PEEK to increase early implant osseointegration. Chemical modifications were also used to improve surface roughness and wettability of the PEEK material [45]. On unmodified PEEK wettability, the water contact angles are generally depicted from 90 to 100 degrees, which are high hydrophobic values [46,47,48]. Here, our modified PEEK surface presented enhanced hydrophilicity with a water contact angle of 90.3° (Table 1), which most likely increased cellular migration and proliferation (Figure 3 and Figure 4). A dental abutment surface can positively influence its interaction with the surrounding tissues by increased biomaterial wettability [30,49,50,51,52,53]. Additionally, our findings showed that PEEK had no disadvantageous effects on cell viability (Figure 1), which previous studies comparing unmodified and modified PEEK [52,53,54] also found to be true. Further, our results showed that modified PEEK greatly improved cell adhesion and migration (Figure 2 and Figure 3), which corresponds with existing literature [55,56,57]. Nevertheless, direct comparison with the current results may be troublesome because of variations in cell cultures times and types, as many studies mainly evaluated osteoblast response on PEEK surfaces.

This study has taken a step in the direction of determining, at a cellular level, the relationship between epithelial gingival tissue and interface of PEEK abutments. There is limited information from in vitro studies assessing the soft tissue response to abutments using different abutment material chemistry. To our knowledge, the present study is the first to focus on assessing the effect of modified PEEK material commonly used for implant abutment on the epithelial soft tissue interface and may have important implications in the clinical practice. Clinical studies have shown that a proper healing process around the abutments is also dependent on the environmental inflammatory conditions. One limitation of the current study may be the non-consideration of inflammation and bacterial endotoxins over the abutment surfaces. Inflammation and bacterial endotoxins are considered part of the normal oral condition and their presence may highlight limited epithelial cell response due to an acute inflammatory cytokine production. Hence, future cell culture studies should be conducted to comprehensively determine the effect of inflammatory conditions on the relationship between soft tissues and biomaterial interface. In addition, although various reports reinforce the importance of modified of PEEK surfaces on increased osteoblast response, in vitro studies, especially for implant abutment on epithelial cells, are still necessary. 

## 5. Conclusions 

Taken together, the findings of the current in vitro study demonstrate that:
-Modified PEEK surface may augment biocompatibility by having a positive impact on viability, adhesion, migration and proliferation of human gingival epithelial keratinocytes as compared to titanium and zirconia. -Accumulation of knowledge on surface abutment effect on surrounding epithelial tissue might provide new insights into the development of future novel nanostructured and biocompatible implants abutments.

## Figures and Tables

**Figure 1 materials-12-01401-f001:**
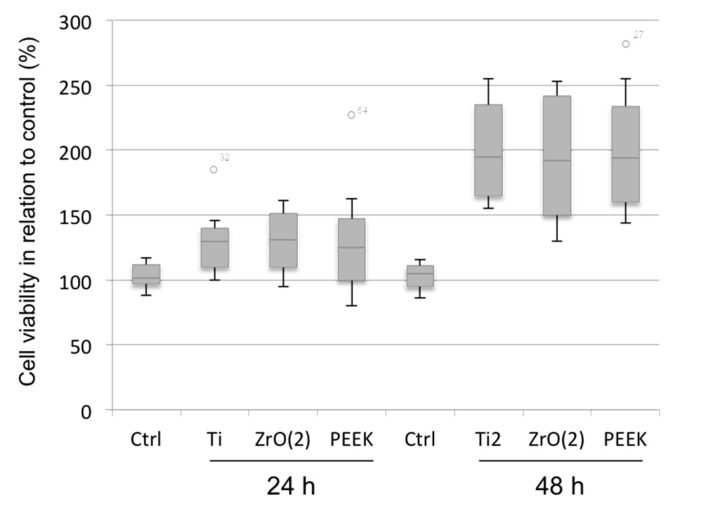
Cell viability assay analysis (MTT) of HGEK 24 h and 48 h compared the control (acrylic polystyrene), titanium (Ti), zirconia [ZrO(_2_)] and PEEK (PEEK) discs. X-axis = points of measurement, Y-axis = OD (optical density).

**Figure 2 materials-12-01401-f002:**
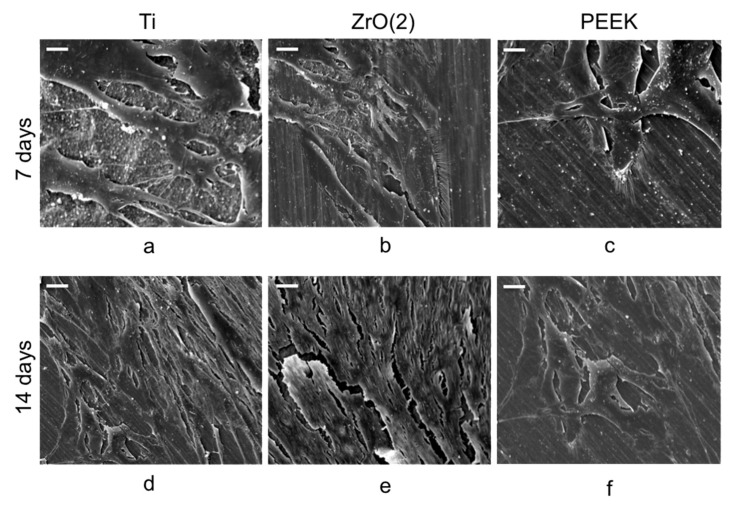
Images of SEM (1k× magnification) obtained from HGEK grown on the disks (confluence reached after seven days) (**a**–**c**). Monolayer formation was found to be continuous up up to 14 days (**d**–**f**). Scale Bar = 10 μm.

**Figure 3 materials-12-01401-f003:**
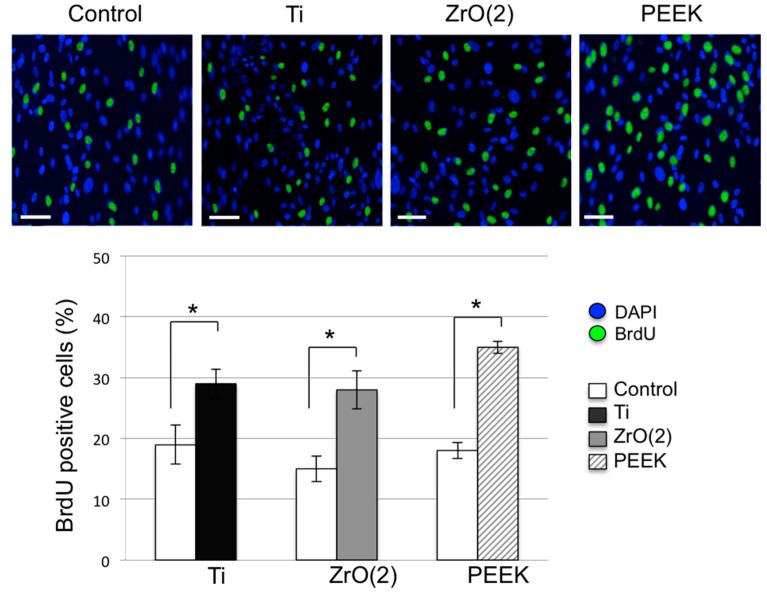
BrdU results (means ± SD, n = 3) represent cellular BrdU incorporation into DNA in arbitrary units after seven days of growth over group disc surfaces compared with control disc. Scale bar: 50 μm, * *p* < 0.05.

**Figure 4 materials-12-01401-f004:**
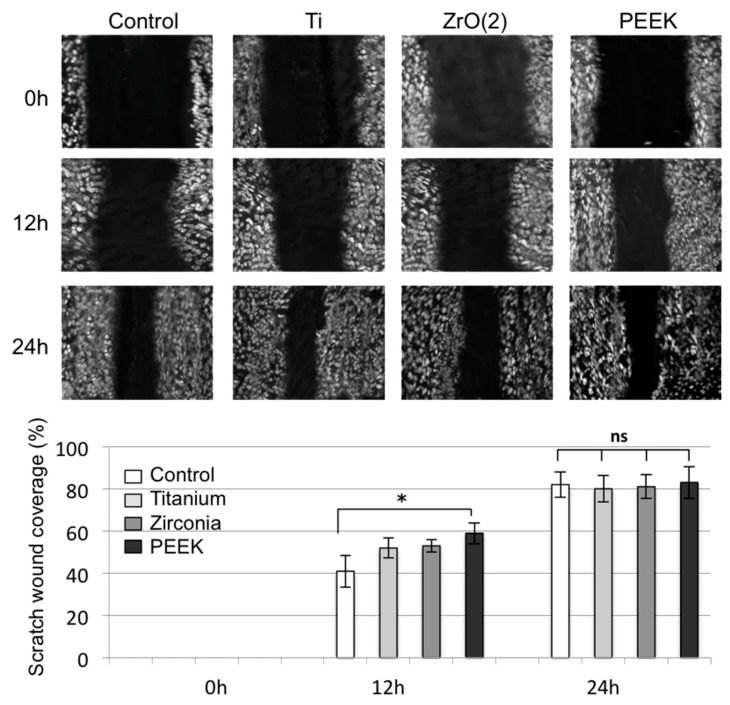
Cell migration (scratch wound healing assay). HEGK cell nuclei were labeled with fluorescent dye Hoechts 33342 and images were recorded 24 h after wounding. Representative images are shown from three independent experiments and dark define the areas lacking cells (wound area, ImageJ). Values of percentage wound closure ± SEM (n = 3): * *p* < 0.01.

**Figure 5 materials-12-01401-f005:**
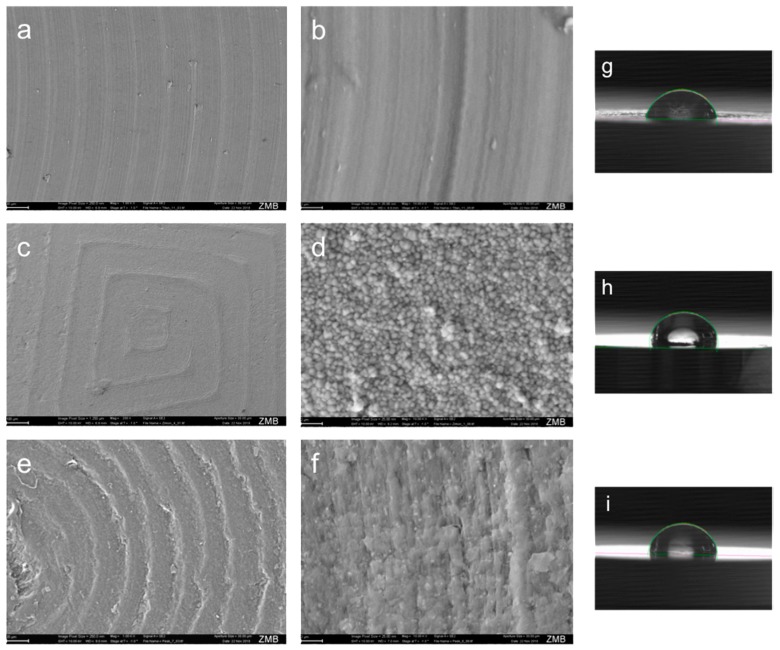
Microphotographs of surface topographies captured by scanning electron microscopy (SEM) at 1k× and 100k× magnification. (**a**,**b**) titanium, (**c**,**d**) zirconia, (**e**,**f**) PEEK. Contact angle of water droplet (1 μL) on (**g**) titanium, H, Zirconia and (**i**) PEEK.

**Table 1 materials-12-01401-t001:** Arithmetic average of surface roughness R_a_ (means and standard deviations [μm]) and wettability (means and standard deviations [°]) of the three tested abutment materials.

Abutment Material	Roughness	R_a_ [μm]	Wettability	[°]
Ti	Smooth	0.086 ± 0.006	Hydrophilic	72.3 ± 5.4
ZrO_2_	Rough	1.152 ± 0.186	Hydrophobic	98.2 ± 8.6
PEEK	Medium	0.827 ± 0.012	Hydrophobic	90.3 ± 7.4

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
