# Peer review of "In Vitro Effect of Modified Polyetheretherketone (PEEK) Implant Abutments on Human Gingival Epithelial Keratinocytes Migration and Proliferation"

_materials, 2019, doi:10.3390/ma12091401_

Round 1
Reviewer 1 Report
The aims were stated well. There was a well written introduction. Materials and methods were clear as well as data analysis. Discussion was well written. I enjoyed the paper and do not have any objections or questions.
Author Response
Cover letter - Response to reviewers |
Subject: Submission of revised article
Title: "In vitro effect of modified polyetheretherketone (PEEK) implant abutments on human gingival
epithelial keratinocytes migration and proliferation"
Manuscript ID: materials-495859
Dear Reviewer,
Thank you for your email dated 24. April 2019 enclosing the link for the reviewer's comments. We appreciate your efforts to evaluate our manuscript and providing us with the opportunity to make the following revisions. The points, which have been raised, are essential and we are thankful for the deliberate screening of the Journal before entering the review process. Our responses are given in a point-by-point way below. Changes to the manuscript are highlighted using the "Track Changes" function in Microsoft Word in the manuscript.
We hope the revised version is now suitable for publication and look forward to hearing from you in due course. Thank you for considering the revised version.
Sincerely,
Liza Ramenzoni, PhD
Clinic of Preventive dentistry, Periodontology and Cariology
Center of Dental Medicine
University of Zurich
Response to Reviewer 1 Comments
Point 1: The aims were stated well. There was a well written introduction. Materials and methods were clear as well as data analysis. Discussion was well written. I enjoyed the paper and do not have any objections or questions.
Response 1: Thank you very much for reviewing our manuscript. We have now used language editing for the manuscript (proof-reading by Wiley English language editing service) and we would be happy to make any further changes that may be still required in the future. We also would like to thank the reviewer for his/her support on the manuscript. After completion of the suggested language edits and following other reviewer’s recommendations, we believe that the revised manuscript has benefitted from an improvement in the overall presentation and clarity. Your recognition of our work is much appreciated.
Reviewer 2 Report
Dear Authors, conratulations for Your study design and results.
However there are some points to be addressed before the manuscript can be accepted.
Introduction: I would add a small paragraph on the importance of abutment-implant connection to prevent the microbial microleakege https://www.ncbi.nlm.nih.gov/pubmed/25589158
Lines 66-73: therefore is repeated. in addition the pragraphs is too long and the conjuction AND is after a full stop. Rephrase it please.
Materials and Methods
Please provide the city and country of the brand of the product You used, including the type of SEM. References of the cell cultures and protocol of SEM observations are required. If You do not have, please include as supplemental material a scheme of the used protocols of the differente passages, in order to guarantee the repeatibility of the experiment.
Discussion: a point that should be discussed is also the antimicrobial properties of the different types of surfaces such as anatase https://www.ncbi.nlm.nih.gov/pubmed/29288072
https://www.ncbi.nlm.nih.gov/pubmed/20528698 https://www.ncbi.nlm.nih.gov/pubmed/16904738
Author Response
Subject: Submission of revised article
Title: "In vitro effect of modified polyetheretherketone (PEEK) implant abutments on human gingival
epithelial keratinocytes migration and proliferation"
Manuscript ID: materials-495859
Dear Reviewer,
Thank you for your email dated 24. April 2019 enclosing the link for the reviewer's comments. We appreciate your efforts to evaluate our manuscript and providing us with the opportunity to make the following revisions. The points, which have been raised, are essential and we are thankful for the deliberate screening of the Journal before entering the review process. Our responses are given in a point-by-point way below. Changes to the manuscript are highlighted using the "Track Changes" function in Microsoft Word in the manuscript.
We hope the revised version is now suitable for publication and look forward to hearing from you in due course. Thank you for considering the revised version.
Sincerely,
Liza Ramenzoni, PhD
Clinic of Preventive dentistry, Periodontology and Cariology
Center of Dental Medicine
University of Zurich
Response to Reviewer 2 Comments
Point 1: Dear Authors, congratulations for Your study design and results. However there are some points to be addressed before the manuscript can be accepted.
Response 1: Thank you for your review of our manuscript. We have did our best to answer each of your points below. In addition, we have now used language editing for the manuscript (proof-reading by Wiley English language editing service) and we would be happy to make any further changes that may be still required in the future. After completion of the suggested changes, we believe that the revised manuscript has benefitted from an improvement in the overall presentation and clarity. Your recognition of our work is much appreciated.
Point 2: Introduction: I would add a small paragraph on the importance of abutment-implant connection to prevent the microbial microleakage https://www.ncbi.nlm.nih.gov/pubmed/25589158
Response 2: Thank you for your suggestion. The manuscript has been edited in the introduction to elucidate importance of abutment-implant connection to prevent the microbial microleakage. Note that this section is on Lines 39-41.
Point 3: Lines 66-73: therefore is repeated. in addition the paragraph is too long and the conjunction AND is after a full stop. Rephrase it please.
Response 3: Thank you! The inconsistencies have now been corrected and the paragraph was shortened. Please find the modifications on Lines: 89-93.
Point 4: Materials and Methods: Please provide the city and country of the brand of the product You used, including the type of SEM. References of the cell cultures and protocol of SEM observations are required. If You do not have, please include as supplemental material a scheme of the used protocols of the different passages, in order to guarantee the repeatability of the experiment.
Response 4: The materials and methods’ details are now correctly included as suggested and every item of methodology was individually checked including protocol details. This should now be in accordance to requested necessary changes throughout the whole material and methods. Thank you for your comment. Please find the modifications between Lines: 97-427.
Point 5: Discussion: a point that should be discussed is also the antimicrobial properties of the different types of surfaces such as anatase:
https://www.ncbi.nlm.nih.gov/pubmed/29288072
https://www.ncbi.nlm.nih.gov/pubmed/20528698 https://www.ncbi.nlm.nih.gov/pubmed/16904738
Response 5: Thank you. The authors agree that the discussion of antimicrobial properties should be included. As suggested, the literature was updated with references to improve the discussion of the manuscript. New paragraph were added (lines 757-772, ref. n. 33-38).
Reviewer 3 Report
Paper is well done and structured.
The model of the study is well performed and the study could be attractive for the Journal readers.
However some minor concerns should be addressed in order to have a final more strong paper.
Introduction section should be enlarged adding some more recent references about dental implant surfaces and highlighting all the clinical implications of the presented study.
Please add the following recent papers increasing the introdcutions and discussion section:
Cicciù, M.; Fiorillo, L.; Herford, A.S.; Crimi, S.; Bianchi, A.; D’Amico, C.; Laino, L.; Cervino, G. Bioactive Titanium Surfaces: Interactions of Eukaryotic and Prokaryotic Cells of Nano Devices Applied to Dental Practice. Biomedicines 2019, 7, 12.
Author Response
Subject: Submission of revised article
Title: "In vitro effect of modified polyetheretherketone (PEEK) implant abutments on human gingival
epithelial keratinocytes migration and proliferation"
Manuscript ID: materials-495859
Dear Reviewer,
Thank you for your email dated 24. April 2019 enclosing the link for the reviewer's comments. We appreciate your efforts to evaluate our manuscript and providing us with the opportunity to make the following revisions. The points, which have been raised, are essential and we are thankful for the deliberate screening of the Journal before entering the review process. Our responses are given in a point-by-point way below. Changes to the manuscript are highlighted using the "Track Changes" function in Microsoft Word in the manuscript.
We hope the revised version is now suitable for publication and look forward to hearing from you in due course. Thank you for considering the revised version.
Sincerely,
Liza Ramenzoni, PhD
Clinic of Preventive dentistry, Periodontology and Cariology
Center of Dental Medicine
University of Zurich
Response to Reviewer 3 Comments
Point 1: Paper is well done and structured. The model of the study is well performed and the study could be attractive for the Journal readers. However some minor concerns should be addressed in order to have a final more strong paper.
Response 1: Thank you for your review of our manuscript. We have did our best to answer each of your points below. In addition, we have now used language editing for the manuscript (proof-reading by Wiley English language editing service) and we would be happy to make any further changes that may be still required in the future. After completion of the suggested changes, we believe that the revised manuscript has benefitted from an improvement in the overall presentation and clarity. Your recognition of our work is much appreciated.
Point 2: Introduction section should be enlarged adding some more recent references about dental implant surfaces and highlighting all the clinical implications of the presented study. Please add the following recent papers increasing the introductions and discussion section: Cicciù, M.; Fiorillo, L.; Herford, A.S.; Crimi, S.; Bianchi, A.; D’Amico, C.; Laino, L.; Cervino, G. Bioactive Titanium Surfaces: Interactions of Eukaryotic and Prokaryotic Cells of Nano Devices Applied to Dental Practice. Biomedicines 2019, 7, 12.
Response 2: Thank you for your recommendation. The manuscript has been edited in the introduction and discussion to increase more recent literature on dental implant surfaces, besides the clinical implications of the present study. Note that this section is on lines 39-41, 67-84 and 757-772.